# Clinical and Histopathologic Features of a Feline SARS-CoV-2 Infection Model Are Analogous to Acute COVID-19 in Humans

**DOI:** 10.3390/v13081550

**Published:** 2021-08-05

**Authors:** Jennifer M. Rudd, Miruthula Tamil Selvan, Shannon Cowan, Yun-Fan Kao, Cecily C. Midkiff, Sai Narayanan, Akhilesh Ramachandran, Jerry W. Ritchey, Craig A. Miller

**Affiliations:** 1Department of Veterinary Pathobiology, College of Veterinary Medicine, Oklahoma State University, Stillwater, OK 74078, USA; jennifer.rudd@okstate.edu (J.M.R.); mtamils@ostatemail.okstate.edu (M.T.S.); shannon.r.cowan@okstate.edu (S.C.); vertoiseaux@gmail.com (Y.-F.K.); jerry.ritchey@okstate.edu (J.W.R.); 2Division of Comparative Pathology, Tulane National Primate Research Center, Tulane University, Covington, LA 70433, USA; cconerly@tulane.edu; 3Oklahoma Animal Disease Diagnostic Laboratory, College of Veterinary Medicine, Oklahoma State University, Stillwater, OK 74078, USA; ssankar@okstate.edu (S.N.); rakhile@okstate.edu (A.R.)

**Keywords:** COVID-19, SARS-CoV-2, domestic cats, feline, animal model, ARDS, pathology

## Abstract

The emergence and ensuing dominance of COVID-19 on the world stage has emphasized the urgency of efficient animal models for the development of therapeutics for and assessment of immune responses to SARS-CoV-2 infection. Shortcomings of current animal models for SARS-CoV-2 include limited lower respiratory disease, divergence from clinical COVID-19 disease, and requirements for host genetic modifications to permit infection. In this study, *n* = 12 specific-pathogen-free domestic cats were infected intratracheally with SARS-CoV-2 to evaluate clinical disease, histopathologic lesions, and viral infection kinetics at 4 and 8 days post-inoculation; *n* = 6 sham-inoculated cats served as controls. Intratracheal inoculation of SARS-CoV-2 produced a significant degree of clinical disease (lethargy, fever, dyspnea, and dry cough) consistent with that observed in the early exudative phase of COVID-19. Pulmonary lesions such as diffuse alveolar damage, hyaline membrane formation, fibrin deposition, and proteinaceous exudates were also observed with SARS-CoV-2 infection, replicating lesions identified in people hospitalized with ARDS from COVID-19. A significant correlation was observed between the degree of clinical disease identified in infected cats and pulmonary lesions. Viral loads and ACE2 expression were also quantified in nasal turbinates, distal trachea, lungs, and other organs. Results of this study validate a feline model for SARS-CoV-2 infection that results in clinical disease and histopathologic lesions consistent with acute COVID-19 in humans, thus encouraging its use for future translational studies.

## 1. Introduction

Since the emergence of severe acute respiratory syndrome coronavirus-2 (SARS-CoV-2) in late 2019, Coronavirus Disease 2019 (COVID-19) has swept across the globe, resulting in over 4 million deaths worldwide as of July 2021 [1]. Although a wide range of clinical symptoms are reported, mortality of COVID-19 patients is closely correlated with progression of viral infection to severe lung disease (pneumonia) and respiratory failure due to acute respiratory distress syndrome (ARDS), which is further complicated by immune cell dyscrasias and hyperinflammation (cytokine storm) in critically ill patients [2,3,4]. Features of pulmonary pathology that are hallmarks of severe COVID-19 (i.e., diffuse alveolar damage with hyaline membrane formation, type II pneumocyte hyperplasia, vascular thrombi, fibrin, and serous exudation) have been difficult to reproduce in animal models, making it impossible to completely understand the pathophysiology of the disease or to test the efficacy of new therapeutics and vaccines [5,6]. Identification of a translational animal model that parallels clinical and pathologic features of disease in addition to route of infection, replication, and transmission kinetics is of paramount importance.

SARS-CoV-2 viral infection and replication within a host requires the presence and distribution of angiotensin-converting enzyme 2 (ACE2) receptors similar to humans [7]. Natural SARS-CoV-2 infections in animals are documented to occur in a diverse range of species, including domestic and exotic cats, dogs, mink, and Golden Syrian hamsters [8,9,10,11], and this diverse host range is largely due to natural expression of ACE2 receptors and host tropism of this receptor with the S protein of SARS-CoV-2 [12,13]. Due to the natural availability of ACE2 receptors and confirmed host susceptibility and transmission [10,14,15,16,17], domestic cats offer an exciting advantage as experimental models for SARS-CoV-2 infection [18,19]. Comorbidities that exacerbate COVID-19 disease, such as hypertension, diabetes, renal disease, and obesity, are readily adapted to feline models [20,21,22,23,24,25]. Furthermore, establishing a SARS-CoV-2-infected feline model is prudent for better understanding zoonotic transmission potential from domestic cats back to people in close contact.

Previous studies have shown successful infection of cats with SARS-CoV-2 via intranasal (1–3.05 × 10^5^ PFU) and/or intra-oral routes (5 × 10^5^ TCID_50_/_mL_) and have confirmed cat-to-cat transmission through both respiratory droplets and aerosolization [16,26,27,28]. However, these studies failed to produce clinical signs in infected cats, and evidence of lower respiratory pathology mirroring severe COVID-19 in humans was not observed [16,26,27,28], potentially due to concentration of the viral inoculum and/or inoculation route. Interestingly, pulmonary disease with diffuse alveolar damage was previously documented in cats intratracheally infected with 1 × 10^6^ TCID_50_ SARS-CoV-1, which also resulted in efficient transmission of virus to uninfected animals [29,30]. Based on outcomes of these former studies, we hypothesized that intratracheal inoculation with a higher concentration of SARS-CoV-2 would increase disease severity in cats due to direct implantation of virus into the lower respiratory tract. Thus, the objective of this study was to test whether inoculation of SARS-CoV-2 via the intratracheal route would result in pulmonary pathology and clinical disease in domestic cats similar to the early exudative phase of COVID-19 in human patients, thereby validating this translational model as a means of investigating the mechanistic foundations of this disease.

## 2. Materials and Methods

### 2.1. Virus

SARS-CoV-2 virus isolate USA-WA1/2020 was obtained from BEI Resources and passaged up to 6 times in Vero (CCL-81, ATCC, Manassas, VA, USA) cells in Vero cell growth medium. Virus stock was titrated and quantified in Vero cells using a standard quantification assay [31]. TCID_50_ was calculated using the Reed and Muench method [32].

### 2.2. Animals

Eighteen (*n* = 18) age-matched (9 months-old) and sex-matched (9 male, 9 female) specific-pathogen-free (SPF) cats were obtained from Marshall BioResources (North Rose, NY, USA). Animals intended for SARS-CoV-2 inoculation were individually housed within Animal Biosafety Level 3 (ABSL-3) barrier animal rooms at Oklahoma State University and fed dry/wet food with access to water ad libitum. Animals intended for sham-inoculation were group-housed within the AAALAC International accredited animal facility at Oklahoma State University. Animals were allowed 30 days for acclimation prior to initiation of the study. Temperature-sensing microchips (Bio Medic Data Systems, Seaford, DE, USA) were implanted subcutaneously in the dorsum after 30 days. Baseline weights, body temperatures, clinical evaluation, and nasal swab sampling were obtained prior to inoculation. All animals were in apparent good health at the onset of the study. (See Institutional Review Board Statement below)

### 2.3. Virus Challenge

Cats were anesthetized with ketamine (4 mg/kg) (Covetrus, Dublin, OH), dexmedetomidine (20 µg/kg) (Orion, Espoo, Finland), and butorphanol (0.4 mg/kg) (Zoetis, Gilrona, Spain) intramuscularly. Cats were positioned in ventral recumbency and intubated so that the end of an endotracheal tube was positioned within the distal trachea as described [33]. In twelve cats, a 3-cc syringe was used to inoculate 1 mL 1.26 × 10^6^ TCID_50_ per mL SARS-CoV-2 isolate USA-WA1/2020 in Dulbecco’s Modified Eagle Medium (DMEM, Gibco, Carlsbad, CA), followed by 2 mL of air from an empty syringe. The remaining six cats were sham-inoculated using sterile PBS via the same method. Viral inoculum dosage was confirmed via virus back-titration of Vero/cells immediately following inoculation.

### 2.4. Clinical Observations and Scoring

Animals were monitored at least once daily for evidence of morbidity and discomfort by a licensed veterinary practitioner. Body weights and temperatures (acquired via thermal microchips) were documented daily for the duration of the study. Full clinical scoring included evaluation of body weight, body temperature, activity levels, behavior, respiratory effort, evidence of ocular/nasal discharge, and recognition of coughing or wheezing. Each factor was assigned a score of 0 (normal), 1 (mild-moderate), or 2 (severe) as described in Table 1. Each clinical factor parameter was added to assign an individual animal a summed clinical score every 24 h for the duration of the study. Cats were observed at rest for respiration rate, activity level, and other notable clinical signs before stimulation.

### 2.5. Sampling

Blood and nasal swab samples were collected under sedation (described above) from all study animals (*n* = 18) at day 0 to serve as a baseline. Blood samples (6 mL) were obtained from all cats via cephalic or medial saphenous venipuncture and processed for viral quantification. Nasal swab samples obtained from the nares of all cats were placed in 2 mL microcentrifuge tubes containing RNAlater solution (Sigma, St. Louis, MO, USA) and stored at −80 °C until processed. At day 4 and day 8 post-inoculation, a subset of SARS-CoV-2-infected cats (*n* = 6 per time point) and sham-inoculated control cats (*n* = 3 per time point) were anesthetized for blood and nasal swab collection, and then humanely euthanized (pentobarbital > 80 mg/kg) and necropsied to collect tissue samples.

### 2.6. Histopathology

Necropsy was performed on six (*n* = 6) SARS-CoV-2-infected cats at 4 dpi and the remaining six (*n* = 6) SARS-CoV-2-infected cats at 8 dpi. Three (*n* = 3) sham-inoculated cats were necropsied at each time point (4 dpi and 8 dpi) to provide control samples. Tissue collection included: lung, tracheobronchial lymph node, nasal turbinates, distal trachea, and kidney. Necropsy tissues were halved and then either placed into 1 mL tubes and frozen at −80 °C or placed into standard tissue cassettes that were then fixed in 10% neutral-buffered formalin for 96 h prior to transferring to 70% ethanol for 72 h. Tissues were then trimmed and processed for histology. Paraffin sections (5 µm thick) were mounted onto positively charged slides, and one slide of each tissue was stained with hematoxylin and eosin (H&E) for microscopic evaluation. Necropsy tissues were evaluated for evidence of inflammation and/or aberrations in lymphoid populations as reported in human COVID-19 patients [34,35,36,37]. Lung tissues were specifically evaluated for the following pathology: alveolar damage (pneumocyte necrosis, hyaline membrane formation), alveolar fibrin deposition (±organization), serous exudate/edema, perivascular infiltrates, alveolar histiocytes, type II pneumocyte hyperplasia, syncytia, thrombosis, and fibrinoid vasculitis. All tissues were assigned a quantitative histologic score based on previously documented criteria [38,39]: 0 = no apparent pathology/change; 1 = minimal change (minimally increased numbers of inflammatory cells); 2 = mild change (mild inflammatory infiltrates, alveolar damage/necrosis, fibrin deposition, and/or exudation); 3 = moderate change (as previously described, but more extensive); 4 = marked changes (as previously described, but with severe inflammation, alveolar damage, hyaline membrane formation, necrosis, exudation, vasculitis, and/or thrombosis). All tissues were evaluated and scored by a board-certified veterinary pathologist blinded to study groups to ensure scientific rigor and reproducibility.

### 2.7. Viral RNA Analysis

Viral RNA analysis was performed on samples from nasal swabs, collected plasma, and tissues. Nasal swabs were immediately broken off into 1.5 mL microcentrifuge tubes containing 200 µL RNAlater Solution (Sigma, St. Louis, MO, USA) and stored at −20 °C. The nasal swabs were vortexed for 15 s and then inverted and centrifuged at 1500 rpm for 10 min. RNA was extracted from frozen necropsy tissues using a QIAamp Viral RNA Mini Kit (Qiagen, Germantown, MD, USA) and tissue homogenizer. SARS-CoV-2 viral RNA was quantified by droplet digital PCR (ddPCR) as previously described [40]. Briefly, ddPCR was performed according to manufacturer’s instructions for the 2019-nCoV CDC ddPCR Triplex Probe Assay (Bio-Rad, Hercules, CA, USA). PCR reaction mixtures were as follows: 5 μL One-Step RT-ddPCR Advanced Kit for Probes Supermix (no dUTPs) (Bio-Rad, Hercules, CA, USA); 2 μL reverse transcriptase; 1 μL 300 mM Dithiothreitol (DTT); 1 μL triplex probe assay for N1, N2, and RPP30 detection (Bio-Rad); 2 μL RNase-free water; and 9 μL RNA template. Duplicate 20 μL samples were partitioned using a QX200 droplet generator (Bio-Rad) and then transferred to a 96-well plate and sealed. Samples were processed in a C1000 Touch Thermal Cycler (Bio-Rad) under the following cycling protocol: 50 °C for 60 min for reverse transcription, 95 °C for 10 min for enzyme activation, 94 °C for 30 s for denaturation, 55 °C for 60 s for annealing/extension for 45 cycles, 98 °C for 10 min for enzyme deactivation, and 4 °C for 30 min for droplet stabilization, followed by an indefinite 4 °C hold. The amplified samples were read in the FAM and HEX channels using the QX200 reader (Bio-Rad). Each experiment was performed with a negative control (no template control, NTC) and a positive control (RNA extracted from SARS-CoV-2 viral stock and diluted 1:12,000). Data were analyzed using QuantaSoft™ Software (Bio-Rad) and expressed as Log_10_ (copies/mL).

### 2.8. Feline ACE2 Analysis

Feline angiotensin-converting enzyme 2 (fACE2) RNA was quantified by ddPCR using methods similar to the above assay for CoV. RNA was extracted from frozen necropsy tissues as outlined above. cDNA was synthesized as previously published [39]. Design of primers and probe targeting fACE2 was developed according to manufacturer’s recommendation, namely keeping GC content between 50–60% for primers and 30–80% for probes and melting temperatures between 50–65 °C for primers and 3–10 °C higher for probes. Oligos were synthesized by Integrated DNA Technologies (IDT, Coralville, IA, USA). The sequences are as follows: Forward: 5′-ACGGAGGCGTAAGGATTT-3′, Reverse: 5′-GTGTGGTAGTGGTTGGTATTG-3′, probe: 5′-CGGATCAGAAATCGAAGG-AAGAA-3′. BLAST analysis [41] of the primer and probe sequences against the domestic cat (*Felis catus*) genome was performed to ensure no similar sequences could be amplified. ddPCR reactions were prepared by adding 10 μL Supermix for Probes (no dUTP) (Bio-Rad), 1 μL of primer/probe mix (final concentration is 500 nM for primers and 250 nM for probe), and 8 μL of cDNA template containing 100 ng RNA equivalent. Droplets were partitioned and PCR executed as above using the following cycling conditions: 95 °C for 10 min, 95 °C for 30 s for denaturation, 58.8 °C for 60 s for annealing/extension for 45 cycles, and 98 °C for 10 min for enzyme deactivation. Droplets were read and analyzed as described above.

### 2.9. Immunohistochemistry

5 µm sections of formalin-fixed, paraffin-embedded lung were mounted on charged glass slides, baked for one hour at 60 °C, and passed through Xylene, graded ethanol, and double-distilled water to remove paraffin and rehydrate tissue sections. A microwave was used for heat-induced epitope retrieval. Slides were heated in a high-pH solution (H-3301, Vector Labs, Burlingame, CA, USA), rinsed in hot water, and transferred to a heated low-pH solution (H-3300, Vector Labs) where they were allowed to cool to room temperature. Sections were washed in a solution of phosphate-buffered saline and fish gelatin (G7765 Sigma, Burlington, MA, USA) (PBS-FSG) and transferred to a humidified chamber for staining at room temperature. Tissues were blocked with 10% normal goat serum (NGS) for 40 min, followed by a 60 min incubation with a guinea pig anti-SARS antibody (NR-10361, BEI, Manassas, VA, USA) diluted 1:1000 in NGS. Slides were washed and transferred to the humidified chamber for a 40 min incubation with a goat anti-guinea-pig secondary antibody (A11073, Invitrogen, Carlsbad, CA, USA) tagged with Alexa Fluor 488 and diluted 1:1000 in NGS. Following washes, DAPI (4′,6-diamidino-2-phenylindole) was used to label the nuclei of each section. Slides were mounted using a homemade anti-quenching mounting medium containing Mowiol (475904, Millipore, Burlington, MA, USA) and DABCO (D2522, Sigma, Burlington, MA, USA) and imaged at 20× with a Zeiss Axio Slide Scanner (Zeiss, White Plains, NY, USA).

### 2.10. Genomic Sequencing

cDNA was synthesized from RNA extracted from viral stock and tracheobronchial lymph nodes of an infected cat. ARTIC [42] V3 primers were used to generate overlapping segments of the viral genome from the cDNA (https://github.com/artic-network/artic-ncov2019/tree/master/primer_schemes/nCoV-2019/V3) (Accessed 3 August 2021). DNA libraries were then prepared using Ligation Sequencing Kit (SQK-LSK-109, Oxford Nanopore Technologies, UK). Library cleanup (AMPure XP, Beckmann Coulter, CA, USA), adapter ligation, and barcoding (EXP-NBD-104, Oxford Nanopore Technologies, UK) were performed following kit recommendations. Libraries were then pooled and sequenced using Oxford Nanopore Technologies sequencing platform following manufacturer recommendations. A mean Q-score of 7 was used for sequencing.

### 2.11. Genome Assembly

De novo sequence assembly was performed using Canu [43]. Reference-assisted consensus genomes were generated from de novo assemblies using SARS-CoV-2 reference genomes Wuhan Hu-1 (GenBank ID: MN908947.3, accessed 3 August 2021) and SARS-CoV-2 USA/WA-CDC-WA1 (GenBank ID: MN985325.1, accessed 3 August 2021). Reference-assisted assembly was performed using samtools [44], minimap2 [45], and Nanopolish [46].

### 2.12. Statistical Analyses

When applicable, data were expressed as means ± SEM and statistically analyzed using GraphPad Prism 9.0 software (La Jolla, CA, USA). Kruskal–Wallis test, Pearson correlations, and ANOVA were used to compare differences in clinical score, histopathology, SARS-CoV-2 viral load, and ACE2 RNA among uninfected and SARS-CoV-2-infected individuals, between sample type, for each tissue individually, and between tissues. For all significant results, pair-wise comparisons were made by post-hoc analysis. *p*-values < 0.05 were considered significant.

## 3. Results

### 3.1. SARS-CoV-2-Infected Cats Exhibit Clinical Signs of Lower Respiratory Disease

In order to clinically assess the feline model in Animal Biosafety Level-3 conditions, a clinical scoring system for feline respiratory disease was developed by integrating features of previously utilized systems [47,48,49] (Table 1). SARS-CoV-2-infected cats exhibited a significant increase in clinical disease scores starting at 4 days post-inoculation (dpi) and then at 5, 6, and 8 dpi when compared to sham-inoculated controls (Figure 1A).

Clinical disease peaked at 4 dpi and continued through to the study endpoint at day 8. The most prominent clinical signs noted were lethargy and increased respiratory effort, which were observed in 100% (12/12) of SARS-CoV-2-infected cats during this study. Both lethargy and respiratory effort increased significantly between 3 and 4 dpi (*p* = 0.0027; *p* = 0.0027) and remained elevated with significantly higher scores through to 8 dpi when compared with day 0 (Figure 1B). Coughing was noted in 4 of 12 infected cats (Appendix A) with peak clinical signs occurring at 4 dpi. Pyrexia (temperature > 39.2 °C) was documented in 8 of 12 SARS-CoV-2-infected cats over the course of the study, while 7 infected cats displayed altered behavior (reduced interest in food or attention) and 5 had measurable weight loss. No cats had ocular or nasal discharge (Appendix A). Sham-inoculated cats did not exhibit clinical signs except for one cat with mild weight loss at 4 dpi.

### 3.2. Feline SARS-CoV-2 Infection Pathology Mirrors Acute COVID-19

Complete post-mortem evaluation was performed for all sham-inoculated (*n* = 6) and SARS-CoV-2-infected cats euthanized on day 4 (*n* = 6) and day 8 (*n* = 6) post-inoculation. Necropsy tissues from SARS-CoV-2-infected cats (lung, trachea, nasal turbinates, and tracheobronchial lymph node (TBLN)) were grossly examined and compared to those from sham-inoculated cats (Figure 2A–C). At 4 dpi, the lungs of SARS-CoV-2-infected cats were grossly heavy and wet, with large multifocal to coalescing regions of dark red consolidation that exuded moderate amounts of edema on cut section (Figure 2B). Gross lung lesions were similar at 8 dpi in SARS-CoV-2-infected cats, though the degree of pulmonary edema was moderately more pronounced (Figure 2C). The TBLN of all SARS-CoV-2-infected cats were diffusely enlarged to 4–5 times normal size at both 4 dpi (*n* = 6) and 8 dpi (*n* = 6).

Microscopic evaluation of selected necropsy tissues (lung, trachea, nasal turbinates, TBLN, and kidney) was performed for all study animals. Tissue sections from all sham-inoculated animals (*n* = 6) were histologically unremarkable (Figure 2D and Appendix A). In contrast, histopathologic features of feline SARS-CoV-2 infection exhibited striking similarities to documented pathologic features of the acute (exudative) and organizing phases of human COVID-19 [34,35,36,37]. At 4 dpi, 100% (6/6) of SARS-CoV-2 -infected cats exhibited a significant degree of lung pathology (interaction, *p* < 0.0001) and prominent histologic features consistent with diffuse alveolar damage (DAD) (Figure 2E,F). Pulmonary edema (5/6 cats), multifocal alveolar damage and necrosis (5/6 cats), perivascular lymphocytic and neutrophilic infiltrates (6/6 cats), and intra-alveolar macrophages (5/6 cats) were significantly elevated in SARS-CoV-2-infected cats at 4 dpi (Appendix A). These changes were occasionally accompanied by multifocal areas of hyaline membrane formation (3/6 cats), mild to moderate amounts of intra-alveolar fibrin (2/6 cats), type II pneumocyte hyperplasia (2/6 cats), and intra-alveolar syncytial cells (2/6 cats). One SARS-CoV-2-infected cat exhibited severe inflammation in the distal trachea at 4 dpi characterized by multifocal areas of submucosal necrosis and fibrinoid vasculitis with multifocal areas of mucosal ulceration and diphtheritic membrane formation (Figure 2G and Appendix A).

Similar histologic features of DAD were also observed in the lungs of SARS-CoV-2-infected cats at 8 dpi. However, the overall pattern of lung injury exhibited more prominent features of vascular injury compared to day 4 (Figure 2H). A significant degree of pulmonary edema/exudate, perivascular inflammatory infiltration, and alveolar histiocytosis was present in 100% of SARS-CoV-2 animals (6/6 cats) at 8 dpi (Appendix A). Alveolar damage and necrosis (4/6 cats) and intra-alveolar fibrin (3/6 cats) were also prominent features at this time point. Moreover, histologic evidence of fibrinoid vasculitis (2/6 cats) and vascular thrombosis (2/6 cats) was also observed at 8 dpi, in addition to occasional viral syncytia (1/6 cats) (Figure 2H). In two of these cats, the tracheal submucosa was multifocally effaced by severe lymphoplasmacytic, histiocytic, and neutrophilic inflammation with necrosis (Figure 2I and Appendix A). Mild, multifocal lymphofollicular inflammation was observed in the nasal turbinates of all SARS-CoV-2-infected cats (6/6) at 4 dpi and in 4/6 cats at 8 dpi, with variable neutrophilic infiltration (Appendix A). All SARS-CoV-2-infected animals (12/12) exhibited mildly increased lymphoid hyperplasia in TBLN at 4 and 8 dpi characterized by increased medullary cords and extranodal proliferations (Appendix A).

Fluorescent immunohistochemistry was performed to detect SARS-CoV-2 positive cells in lungs and TBLN of 2 SARS-CoV-2-infected cats (*n* = 1 at 4 dpi, *n* = 1 at 8 dpi). At both time points, low numbers of mononuclear cells positive for SARS-CoV-2 nucleoprotein were detected within the medulla of the TBLN. However, no positive cells were observed in lungs of these animals. (Figure 3). A positive linear correlation exists between peak clinical scores and histopathology scores of the lungs in SARS-CoV-2-infected cats (*p* = 0.0002; R^2^ = 0.5884) indicating that severe clinical signs of disease correlate with pulmonary pathology (Appendix A). No significant histopathologic findings were observed in renal tissues at either time point.

### 3.3. Viral Sequencing and Genome Assembly

A total of 1,118,745 reads from an infected tracheobronchial lymph node collected at 4 dpi and a total of 895,981 reads from viral stock, with a genome coverage of more than 10,000×, were generated after length filtering of the sequence outputs. Consensus genome assemblies from SARS-CoV-2 USA-WA1/2020 viral stock and the tracheobronchial lymph node were compared with each other. The genomes were mostly identical, except at nucleotide positions 8782 and 18,060 of ORF 1ab; nucleotide position 22,296 of S gene; and nucleotide position 28,144 of ORF8 (Table 2). Nucleotide changes in ORF1ab resulted in synonymous mutations. Nonsynonymous mutations resulting in amino-acid change from arginine to histidine at nucleotide location 22,296 (R245H) in S gene and from serine to leucine at nucleotide position 28,144 (S251L) in ORF8 were identified.

### 3.4. ACE2 Expression and Viral RNA in Feline Tissues during SARS-CoV-2 Infection

SARS-CoV-2 viral RNA and fACE2 RNA expression were quantified in the nasal turbinates, TBLN, distal trachea, kidneys, and lungs of all SARS-CoV-2-infected cats (*n* = 12) and sham-inoculated controls (*n* = 6) using ddPCR (Appendix A). Viral RNA was detected in 100% of tissues collected at 4 dpi from SARS-CoV-2-infected cats (Figure 4A). At 8 dpi, viral RNA was also detectable in the lung, TBLN, and kidney tissues of all (6/6) infected cats, in 5/6 cats in the nasal turbinates, and in 5/6 cats in the distal trachea. No SARS-CoV-2 viral RNA was detected in tissues collected from sham-inoculated cats at either time point (Appendix A). SARS-CoV-2 viral RNA copies were elevated in the TBLN at 8 dpi compared with day 4 samples, although this trend was not significant (*p* = 0.0567). In contrast, SARS-CoV-2 viral load in lung samples was significantly lower at 8 dpi than at 4 dpi (*p* = 0.0007) (Figure 4A). A positive linear correlation was observed between SARS-CoV-2 RNA in the lung and pulmonary histopathology scores of SARS-CoV-2-infected cats (*p* = 0.0183; R^2^ = 0. 3012). SARS-CoV-2 RNA was not reliably detected in nasal swabs or plasma of infected cats at either time point.

In sham-inoculated cats, Kruskal–Wallis test revealed that fACE2 RNA in the nasal turbinates was significantly higher than in the lungs (*p* = 0.0093) and TBLN (*p* = 0.0049) (Appendix A). fACE2 RNA was also higher in the kidney when compared to the lungs (*p* = 0.0003), trachea (*p* = 0.0034), and TBLN (*p* = 0.0001). These findings were similar in SARS-CoV-2-infected animals at 4 and 8 dpi, with fACE2 RNA levels being significantly higher in the nasal turbinates and kidney versus other tissues (*p* < 0.05) (Appendix A). Overall, fACE2 RNA in the kidney was significantly increased in SARS-CoV-2-infected cats at 4 dpi when compared to both sham-inoculated controls, and SARS-CoV-2-infected cats at 8 dpi (ANOVA, *p* < 0.0001) (Figure 4B). No other significant changes in ACE2 RNA were observed.

## 4. Discussion

The potential of this feline model for future evaluation of COVID-19 is extensive. Challenges with earlier feline models of SARS-CoV-2 infection included a lack of both clinical disease and pathology of the lower respiratory tract that resembled lesions seen in patients with COVID-19. The differences in clinical presentation between previous feline models and the model described here are likely attributable to modifications in routes and concentration of inoculation. In this study, SARS-CoV-2 was inoculated through an intratracheal route and at a higher concentration than previously reported [16,26,27]. Route of inoculation is an important consideration when establishing an animal model for disease, and previous studies have exhibited marked differences in primary disease severity and distribution based on route of inoculation [51,52].

While previous feline models offer value for study of asymptomatic infections, viral shedding, and transmission of SARS-CoV-2, cats infected through an intratracheal route exhibit clinical disease that aligns with that seen in early phases of acute COVID-19. Clinical assessment of infectious lower respiratory disease in a cat can be challenging, and it is not uncommon for cats with confirmed histologic infectious pneumonia to have limited clinical respiratory signs [53]. Therefore, the clinical signs of respiratory disease induced in this model are highly significant. A novel clinical scoring system was designed that could be applied in an Animal Biosafety Level-3 facility to carefully assess for clinical disease. Interestingly, the disease noted in the SARS-CoV-2-infected cats was similar to that described in hospitalized patients with COVID-19. Clinical disease of hospitalized human COVID-19 patients is characterized by fever (70–90%), dry cough (60–86%), shortness of breath (53–80%), and fatigue (38%) [54], while predominant clinical signs in SARS-CoV-2-infected cats consisted of fever, cough, lethargy, and increased respiratory effort, with lethargy and increased respiratory effort being the most notable clinical signs (Figure 1).

In addition to clinical signs of lower respiratory and systemic disease, SARS-CoV-2-infected cats also exhibited conspicuous pulmonary lesions of diffuse alveolar damage (DAD) by 4 dpi, and additional evidence of vascular damage by 8 dpi (Figure 2). Specific histopathological lesions aligned closely with those reported in human COVID-19 patients [34,35,36,37,55,56], including DAD resulting in hyaline membrane formation, type II pneumocyte hyperplasia, occasional intra-alveolar syncytial cells, and the development of fibrinous exudate and vascular thrombi. To the authors’ knowledge, this is the first report of hyaline membrane formation and type II pneumocyte hyperplasia in feline SARS-CoV-2 infection. Peak clinical disease scores positively correlated with severity of histologic lesions in the lungs (Appendix A), which further supports that cats with marked pulmonary histologic damage also had more severe clinical signs of disease.

Surprisingly, intratracheal inoculation of SARS-CoV-2 did not produce high viral RNA loads in the lungs as compared with other studies in which the inoculate was delivered via the intranasal route [26]. However, despite bypassing the upper airway, virus was still detected in the nasal turbinates by 4 and 8 dpi, suggesting the virus may utilize the mucociliary escalator to travel up the respiratory tree and establish infection intranasally even without intranasal inoculation. Although seemingly lower quantities of SARS-CoV-2 RNA were recovered from lungs of intratracheally inoculated cats, the damage to lung tissues was highly evident, indicating that extensive pulmonary damage will occur even without high levels of viral replication within the pulmonary tissue at 4 and 8 dpi. Viral migration from lung to the TBLN occurred quickly (by 4 dpi) and this TBLN involvement is a novel finding for the feline model of SARS-CoV-2, including detection of viral antigen within the TBLN via fluorescent immunohistochemistry. Sequencing of viral RNA from infected TBLN did not greatly differ from that of the viral inoculum, suggesting that genetic medication is not required for SARS-CoV-2 infection in felids, although further studies evaluating the effect of these few mutations on infectivity or pathogenicity are needed.

Similar to humans, fACE2 RNA expression varied by tissue location, but was relatively low in the lungs of both infected and uninfected cats. It is important to note that RNA measurements indicate an upregulation or downregulation of production of proteins, but do not necessarily indicate an absolute number of receptors available. However, it is possible that inefficient replication and rapid clearance of SARS-CoV-2 in the lungs is related to lower expression of ACE2 receptors as compared with nasal turbinate ACE2. Histopathology shows that cells regularly expressing ACE2 are damaged in the lung, and this viral-induced pulmonary epithelial pathology may contribute further. ACE2 RNA copies in the feline kidney are significantly higher than that of other assessed tissues, and viral infection resulted in a significant upregulation of ACE2 RNA by 4 dpi and then a subsequent reduction by 8 dpi. Hypertension and activation of the renin–angiotensin system may have driven this rise in ACE2 in order to counterbalance system effects of infection, and future studies should include blood pressure evaluation in conjunction with other clinical parameters such as oxygen saturation, chemistry panels, and imaging. Further studies are needed to fully understand the role of ACE2 in SARS-CoV-2 viral replication kinetics and disease.

Limitations to this study include sample sizes as well as sampling time points. Further studies are needed to try and better identify peak viral loads in various tissues as well as ACE2 expression in order to investigate when viral clearance occurs and when this feline model moves from early exudative disease to more organized, fibrotic disease, which would require an extended study beyond 8 days. A better delineation of these events would add the model’s value and potential for use at other stages of disease. In addition, transmission and viral shedding after intratracheal inoculation of SARS-CoV-2 should be evaluated and compared with that of intranasal inoculation and spread. Expansion of in-depth diagnostics was limited due to animal biosafety level requirements and availability of resources, but future studies will seek to evaluate other clinical parameters, such as oxygen saturation, thoracic imaging, complete blood counts, chemistry panels, and urine analysis to better assess damage to other organ systems and further compare with human disease.

The outcomes reported in this study establish the first feline model of SARS-CoV-2 infection with significant lower respiratory disease and features of diffuse alveolar damage and ARDS analogous to those seen in the early exudative phase of human COVID-19. In addition, SARS-CoV-2-infected cats exhibited clinical signs of lower respiratory disease characterized by increased respiratory effort and coughing in addition to signs of systemic involvement such as pyrexia and lethargy. This feline model of SARS-CoV-2 infection offers an animal model that closely mirrors both clinical disease and pathology identified in hospitalized patients with severe COVID-19, making the model a potential option for future studies addressing novel therapeutics for COVID-19. Therapeutic measures can be thoroughly assessed for improvement in pathology and mitigation of clinical disease in cats before being validated in human trials, and more thorough evaluation of the feline immune response to infection may elucidate other options for COVID-19 treatments that could mitigate disease and improve clinical outcomes. The continued emergence of novel variants, circulating globally, ceaselessly contributes to the complexity and duration of this pandemic. While the role of cats in zoonotic transmission is still under investigation, the applicability of a clinically significant SARS-CoV-2 feline model with pathological lesions that mirror severe COVID-19 is of high significance for future studies. This animal model offers ease of use, which can positively impact further vaccination and control strategies necessary to achieve an end to the rapid spread of COVID-19. This model also offers utility in a One Health approach to the role of companion animals in disease transmission, antigenic drift, and more thorough evaluation of the potential for feline contributions to the spread of SARS-CoV-2.

## Figures and Tables

**Figure 1 viruses-13-01550-f001:**
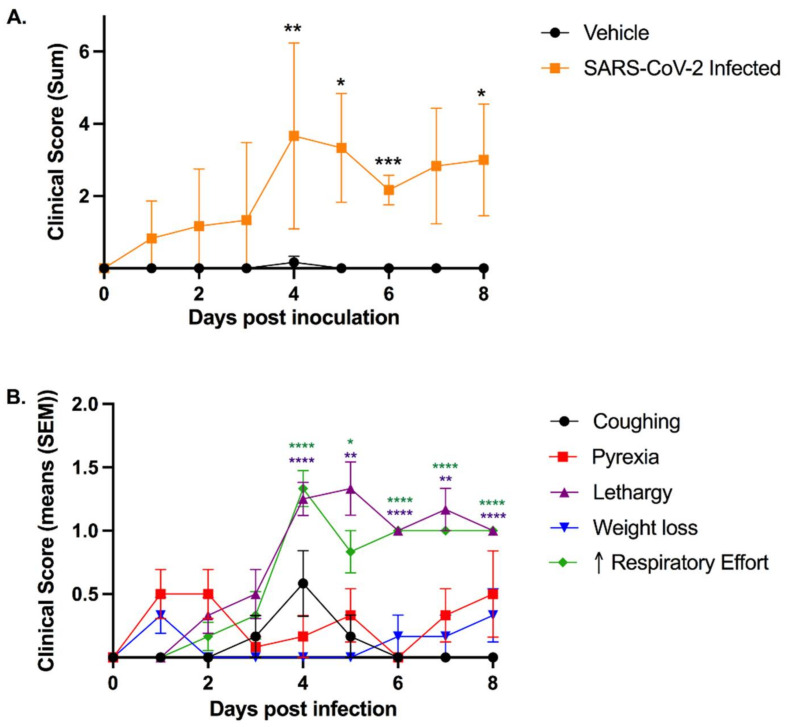
Intratracheal SARS-CoV-2 inoculation results in clinical disease. Clinical parameters were assessed using the feline respiratory disease clinical scoring system (see Table 1). (**A**) Clinical parameters summated to provide an overall clinical score per cat per day. Clinical disease severity peaked at 4 dpi and was significantly higher than sham-inoculated cats at 4 dpi (*p* = 0.0054), 5 dpi (*p* = 0.0257), 6 dpi (*p* = 0.0004), and 8 dpi (*p* = 0.0453). A noticeable trend in severity was also noted at 7 dpi as compared with sham-inoculated controls (*p* = 0.0654). (**B**) Lethargy and increased respiratory effort were the most prominent clinical signs observed in SARS-CoV-2-infected cats, both of which were significantly increased between days 3 and 4 (*p* = 0.0027; *p* = 0.0027) and remained significantly elevated in infected cats after 4 dpi as compared to day 0. Coughing was most prominent at 4 dpi and was identified in 4/12 infected cats. Pyrexia was noted in 8/12 cats over the course of the study. Data are expressed as means ± SEM. Statistical comparisons made via mixed-effects analysis. * *p* < 0.05; ** *p* < 0.01; *** *p* < 0.001; **** *p* < 0.0001.

**Figure 2 viruses-13-01550-f002:**
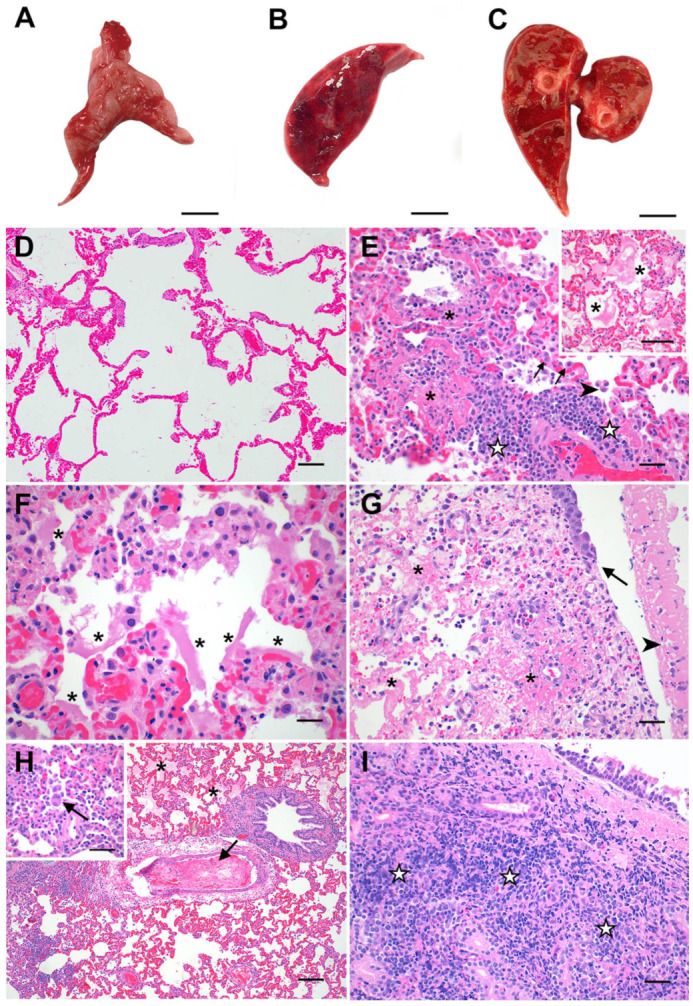
Pathologic features of acute SARS-CoV-2 infection in cats are analogous to the exudative phase of COVID-19. Compared to lungs from healthy sham-inoculated cats (**A**), the lungs of SARS-CoV-2-infected cats were diffusely consolidated, dark red, and edematous at both 4 dpi (**B**) and 8 dpi (**C**). The lungs of healthy, uninfected cats (**D**) were histologically normal, with open alveoli and minimal atelectasis. At 4 dpi, the lungs of SARS-CoV-2-infected cats (**E**) exhibited discrete foci of alveolar inflammation and necrosis with fibrin deposition (✱), increased alveolar macrophages (arrowhead), perivascular lymphocytes (☆), and type II pneumocyte hyperplasia (arrows). The alveoli in these cats’ lungs were frequently filled with large amounts of edema and fibrin strands ((**E**) inset), and there were multifocal areas of hyaline membrane formation (**F**) (✱). The distal trachea of one SARS-CoV-2-infected cat (**G**) was multifocally ulcerated at 4 dpi (arrow) with diphtheritic membrane formation (arrowhead) and multifocal areas of submucosal necrosis and fibrinoid vasculitis (✱). At 8 dpi (**H**), fibrinoid vasculitis, vascular thrombosis (arrow), and occasional syncytial cells (H inset) were observed in addition to the histopathologic changes described above. Tracheal lesions observed at 8 dpi (**I**) were characterized by varying degrees of lymphoplasmacytic, histiocytic, and neutrophilic inflammation, with multifocal areas of submucosal necrosis. Magnification: (**A**–**C**) scale bar = 1 cm; (**D**,**H**) 10×, scale bar = 100 µm; (**E**) 20×, scale bar = 50 µm; ((**E**) inset, (**G**), (**H**) inset, (**I**)) 40×, scale bar = 25 µm; (**F**) 60×, scale bar = 17 µm.

**Figure 3 viruses-13-01550-f003:**
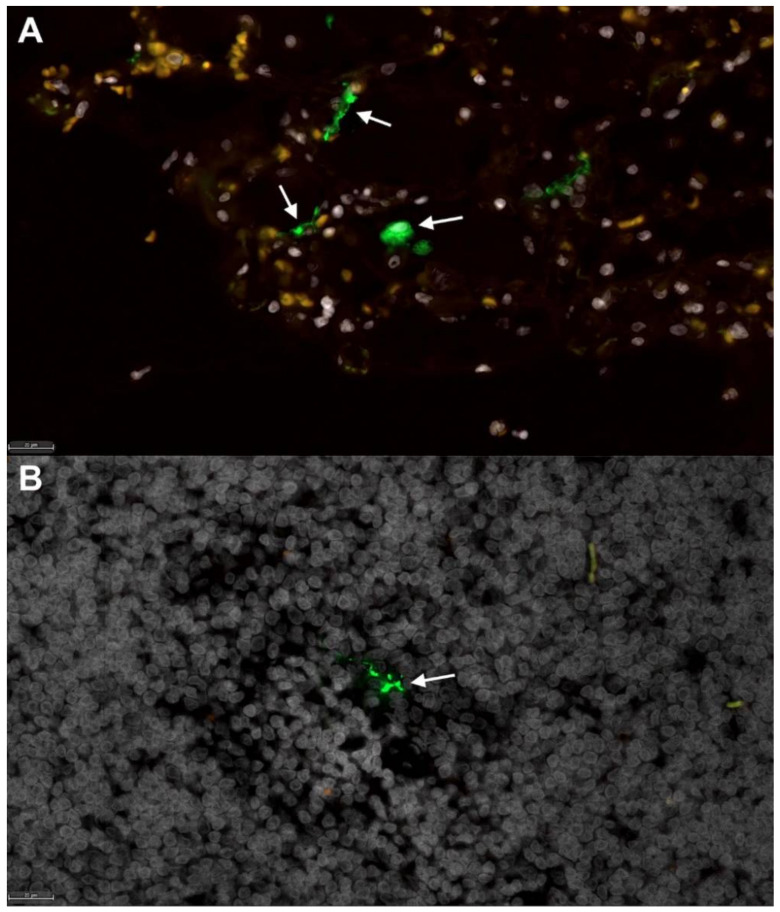
Fluorescent immunohistochemistry for SARS-CoV-2 nucleoprotein identifies mononuclear cells in tracheobronchial lymph nodes of intratracheally infected cats. Low numbers of SARS-CoV-2 positive cells (green, white arrows) are detected in (**A**) positive control tissue (lung) from an African Green Monkey infected with SARS-CoV-2 [50], and within (**B**) mononuclear cells in the TBLN of SARS-CoV-2-infected cats (green, white arrow). White, DAPI/nuclei; green, CoV-2. Magnification 40×, scale bar = 20 µm.

**Figure 4 viruses-13-01550-f004:**
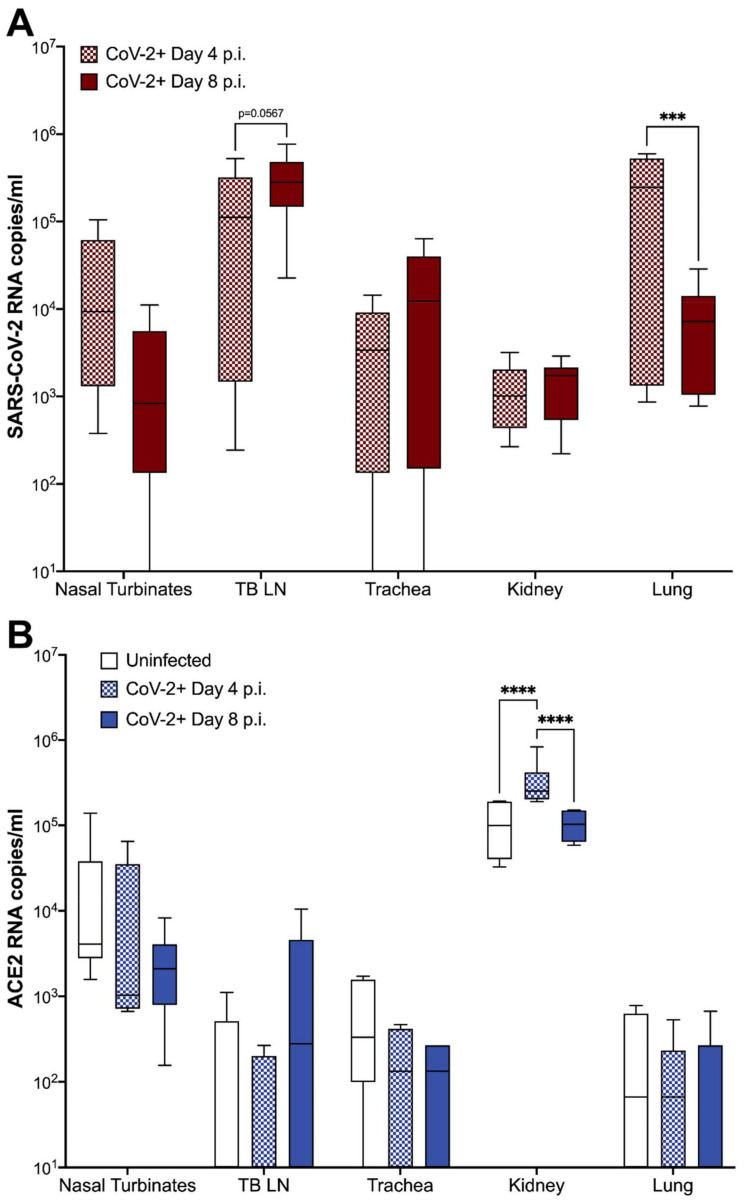
fACE2 RNA and SARS-CoV-2 viral RNA quantification in feline tissues. Extraction of SARS-CoV-2 and ACE2 RNA was performed as described from tissues samples collected at either 4 or 8 dpi. Tissue samples included nasal turbinates, tracheobronchial lymph node (TBLN), trachea, kidney, and lung. (**A**) SARS-CoV-2 RNA copies were detected in all tissues collected from cats inoculated with SARS-CoV-2. No viral RNA was detected in tissues from sham-inoculated cats. In SARS-CoV-2-infected cats, viral RNA copies were slightly increased in the TBLN between 4 to 8 dpi (*p* = 0.0567), while viral RNA load in the lungs significantly decreased over the same period (*p* = 0.0007). (**B**) fACE2 receptor RNA is significantly increased in the kidney of SARS-CoV-2-infected cats at 4 dpi compared to sham-inoculated controls (*p* < 0.0001) SARS-CoV-2-infected cats at 8 dpi (*p* < 0.0001). Data are expressed as means ± SEM. *n* = 6 cats per group. Statistical comparisons made via two-way ANOVA. *** *p* < 0.001; **** *p* < 0.0001.

**Table 1 viruses-13-01550-t001:** Clinical Scoring System for Feline Respiratory Disease. A scoring system was designed to assess clinical lower respiratory and systemic disease in the feline model. Each cat was scored daily at the same time point (morning) by a small animal clinician (JMR). Cats were assigned a score from 0 to 2 for each clinical parameter: body weight, temperature, respiratory effort, activity, behavior, ocular/nasal discharge, coughing/wheezing. The parameter scores were summed to assign an overall score per cat per day. Potential scores could range from 0 (healthy with no signs of disease) to 14 on any given day. Resting respiratory rate was considered normal if <36 breaths per minute. Marked increases in rate were >50 breaths per minute at rest. Temperatures were obtained via thermal microchips. Body weights were obtained last to limit stress affecting clinical scoring.

Clinical Parameter	0 (Healthy)	1	2
Body Weight	No weight loss	0 to 5% weight loss	>5% weight loss
Temperature	37.2 to 39.1 °C	39.2 to 39.7 °C	>39.7 °C
Respiratory Effort	Normal resting respiratory rate and normal effort	Mild tachypnea, but no overt increase in effort	Marked increase in both respiratory rate and effort; dyspnea
Activity	Normal	Reduced activity when disturbed * (lethargy)	Little to no activity disturbed *; reduced activity stimulated **
Behavior	Normal	Reduced interest in food and/or attention	Anorexia and lack of interest
Ocular/Nasal Discharge	None	Mild discharge noted	Discharge evident from both nasal and ocular regions
Coughing/Wheezing	None	Mild wheezing, but no coughing	Coughing and/or marked wheezing

* Disturbed: observer in the room, but kennel unopened. ** Stimulated: kennel open.

**Table 2 viruses-13-01550-t002:** Mutations observed from infected tracheobronchial lymph node when compared with viral inoculum. Genomes were mostly identical except at nucleotide positions 8782 and 18,060 of ORF 1ab; nucleotide position 22,296 of S gene and nucleotide position 28,144 of ORF8.

Nucleotide Change	Gene	Amino Acid Shift
T8782C	ORF1ab	S2839S
T18060C	ORF1ab	L5932L
G22296A	S gene	R245H
C28144T	ORF8	S251L

## Data Availability

All relevant data associated with this study have been deposited in a public repository: https://doi.org/10.6084/m9.figshare.14449773.

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
