# Peer review of "Clinical and Histopathologic Features of a Feline SARS-CoV-2 Infection Model Are Analogous to Acute COVID-19 in Humans"

_viruses, 2021, doi:10.3390/v13081550_

Round 1
Reviewer 1 Report
This manuscript, by Rudd et al, aimed to characterise a feline model of SARS-CoV-2 following intra-tracheal challenge instead of via intra-nasal or intra-oral routes described previously. This manuscript was interesting, well written, and pretty comprehensive.
One thing that did come to mind is the public perception of using cats as a model. Could there be problems with the cat-lovers of the world?
In addition, I have the following minor observations:
- Line 106: CCL81 cells are Vero, not VeroE6. Was this a mixture of cells, or were the two cell lines used at different times?
- Line 148: usually neutral-buffered formalin rather than formaldehyde
- Line 149: the "Five um paraffin sections" was a little unclear - n=5 um scale sections, or sections that are 5um?
- Line 240: GenBank sequence for WA1?
- Section 3.3: Unclear how many lymph nodes were sequenced from; what timepoint was the sample collected from?
- Figure 4 (Line 402): I sincerely hope you mean SARS-CoV-2, I can't deal with a SARS-CoV-3 just yet...!
- Discussion: I liked the limitations and future directions paragraph (Line 473-484). Do you think the animals would survive infection, or would it be a lethal model of infection? Would taking the infection out longer be useful
- References: Some journal names not abbreviated
Reviewer 2 Report
I highly appreciate the idea of this original study and its findings, as well as the researchers’ hard work on the experiments. The manuscript is generally well written, with attention to details and well-structured in most paragraphs. Introduction: very well introducing the importance of the study; background is based on recent and correct references; pertinent and clear. Materials and Methods – well described and in details. Results: impressive, supported by good statistics and well-illustrated with figures of excellent quality (including those in Supplementary file). Tables are correct and nicely synthesizing the data (including Supplementary ones). Discussion paragraph appears well conceived, including limitations and strong points of this original research. Please see my suggestions for minor changes below:
- Abstract:
- The authors wrote: “lesions consistent with severe COVID-19 in humans”. Title mentions “acute”. Is it “acute” or “severe”? It is not the same.
- “Natural ACE2 expression, paired with histopathologic correlates between this model and COVID-19, encourage its use for future translational studies”. Please rephrase for better understanding and emphasizing.
- The Abstract should be re-written in order to be clearer what was performed and what was found. In the current form, sentences are mixed (methods and results). And, then, a firm conclusion should be written (in the present form, it appears in the middle of the Abstract). This study is a very good one and it deserves a much better Abstract, reflecting what the authors found.
- Introduction:
- Line 58: please insert “shown” after “have”: “Previous studies have successfully infected”
- Aim of the study should be emphasized more by the end of Introduction.
- Sentences reporting about outcomes of this study: from line 70 to 74 – do not belong here; they are mentioned in Discussion
- Sentence about applicability of this study: lines 75-78 – do not belong here; they are mentioned in Discussion.
- Results:
- Line 279 – day 4 or day 5? (it appears to be day 5 in Table S1)
